# Advantages of the Utilization of Wide-Field OCT and Wide-Field OCT Angiography in Clinical Practice

**DOI:** 10.3390/diagnostics14030321

**Published:** 2024-02-01

**Authors:** Maciej Gawęcki, Krzysztof Kiciński

**Affiliations:** 1Dobry Wzrok Ophthalmological Clinic, 80-822 Gdansk, Poland; 2Department of Ophthalmology, Pomeranian Hospitals, 84-120 Wejherowo, Poland; krzysztofkg999@icloud.com

**Keywords:** wide-field fluorescein angiography, wide-field optical coherence tomography, wide-field OCT angiography, diabetic retinopathy, retinal vein occlusion, peripheral retinal degeneration

## Abstract

Wide-field (WF) retinal imaging is becoming a standard diagnostic tool for diseases involving the peripheral retina. Technological progress elicited the advent of wide-field optical coherence tomography (WF-OCT) and WF-OCT angiography (WF-OCTA) examinations. This review presents the results of studies that analyzed the implementation of these procedures in clinical practice and refers to them as traditional and ultra-wide-field fluorescein angiography (UWF-FA). A PUBMED search was performed using the terms WF-OCT OR WF-OCTA OR UWF-FA AND the specific clinical entity, and another search for diabetic retinopathy (DR), retinal vein occlusion (RVO), Coats disease, peripheral retinal telangiectasia, peripheral retinal degeneration, lattice degeneration, and posterior vitreous detachment. The analysis only included the studies in which the analyzed field of view for the OCT or OCTA exam was larger than 55 degrees. The evaluation of the extracted studies indicates that WF imaging with OCT and OCTA provides substantial information on retinal disorders involving the peripheral retina. Vascular diseases, such as DR or RVO, can be reliably evaluated using WF-OCTA with results superior to standard-field fluorescein angiography. Nevertheless, UWF-FA provides a larger field of view and still has advantages over WF-OCTA concerning the evaluation of areas of non-perfusion and peripheral neovascularization. Detailed information on the vascular morphology of peripheral changes should be obtained via WF-OCTA and not angiographic examinations. WF-OCT can serve as a valuable tool for the detection and evaluation of vitreoretinal traction, posterior vitreous detachment, and peripheral retinal degeneration, and guide therapeutic decisions on a patient’s eligibility for surgical procedures.

## 1. Introduction

Optical coherence tomography (OCT) has become a standard method of retinal diagnostics for ophthalmologists. Since its commercial introduction at the beginning of this century, OCT has undergone many technological developments that have improved image resolution, artifact elimination, and the depth of scanning. Nevertheless, for many years, the horizontal dimensions of the scan were limited to 6 × 6 mm, that is, to the macular area. Over time, wider scans became possible, such as 12 × 12 mm. These scans covered the area of a classic fundus camera with a 55-degree field and were initially named wide-field OCT (WF-OCT) [1]. The peripheral retina was still difficult to assess, especially when reaching outside the vascular arcades of the posterior pole; it required repositioning the fixation point and creating a mosaic of a few images [2]. Such a montage provided a field of view between 70 and 80 degrees, as reported for the Plex® Elite 9000, (Carl Zeiss Meditec Inc., Dublin, CA, USA) device [3]. Recent advancements in the optics used in OCT and an increase in the scanning speed have enabled wide-range scans of more than 20 mm in width. For example, Xephilio OCT-S1 (Canon, Tokyo, Japan) which is based on a swept-source laser (SS-OCT), can capture a retinal area of 23 × 20 mm, which is approximately equivalent to an 80-degree viewing angle obtained in a single scan. The mosaic of 4–5 images enables the visualization of an area of 31 × 27 mm, thus reaching beyond the vortex veins. Some wide-filed OCT devices have the option of OCT angiography, which promises the reliable evaluation of the peripheral perfusion without the need to perform standard angiography, which requires the injection of dye.

A consensus on defining WF-OCTA was eventually reached by international experts (the Delphi approach). Wide-field OCTA was defined as capturing a field of view of at least 90 degrees [4]. In practice, none of the commercially available OCT devices can obtain such a field of view with a single scan; it is only possible with a mosaic of wide-field images. Such a wide field of scanning is often called ultra-wide-field (UWF) in the medical literature. Sometimes, the term wide field is also used for OCTA scans of 12 × 12 mm. In this review, we analyzed studies that involved fields of view larger than 12 mm in width; in practice, this means areas larger than a 55-degree field of view. However, it must be noted that inconsistencies in the nomenclature of wide-field imaging are still present in the ophthalmological literature. As such, the present analysis of published studies was based on numbers (the dimensions of the scans) rather than the terms used in the studies.

The capabilities of wide-field OCT devices must be analyzed in reference to the color fundus and fluorescein angiography wide-field imaging (UWF-CP and UWF-FA). The International Widefield Imaging Group recommends using the term wide field for images that capture retinal areas posterior to the vortex vein ampulla in all four quadrants and ultra-wide-field for images that show retinal features anterior to the vortex vein ampullae [5]. Such a recommendation sets the division point between WF and UWF imaging at approximately 100 degrees of field of view. Fundus cameras available on the market can capture significantly larger fields of view compared to OCT devices. For example, an Optos scanning laser ophthalmoscopy (SLO) camera provides standard images of 200 degrees with a single scan. The Zeiss Clarus 700 can obtain 133-degree images with a single scan and 200-degree images with a montage of two scans.

The present review aims to provide a descriptive and systematic analysis of the advantages of wide-field OCT and wide-field OCTA examinations compared to established angiographic imaging, such as standard FA or UWF-FA and UWF-CP. The review was performed for the specific clinical entities involving pathologies located at the peripheral retina.

## 2. Material and Methods

The database search was performed for clinical entities for which diagnostics of the peripheral retina are crucial for diagnosis and treatment. The search was performed using the following combination of terms: wide-field OCT OR wide-field OCT angiography OR wide-field fluorescein angiography AND the specific clinical entity. A separate search of the PUBMED database was performed for the following retinal disorders: diabetic retinopathy, retinal vein occlusion, Coats disease, peripheral retinal telangiectasia, peripheral retinal degeneration, lattice degeneration, posterior vitreous detachment, and retinal retinoschisis. The analysis only included the studies in which the analyzed field of view for the OCT or OCTA exam was larger than 55 degrees (areas larger than 12 × 12 mm) on either a single scan or a mosaic of scans. The results of the search are presented in the sections below, which are organized by disease. Rare disorders that were noted in the context of wide-field OCT examinations during the search were analyzed separately and are presented in a dedicated section below.

## 3. Diabetic Retinopathy

Diabetic retinopathy (DR) is a vascular disorder often associated with lesions located at the peripheral retina, which influence its classification, progression, and the development of diabetic macular edema. These issues were analyzed with UWF imaging: CP and FA. The use of UWF-FA enables a more precise classification of DR. Employing UWF-FA often makes the classification of DR more precise (for example, from non-proliferative to proliferative) as areas of neovascularization (NV) not detected with standard exams can be detected with UWF imaging [6,7,8,9]. Moreover, the diagnosis of peripheral lesions in DR, especially peripheral areas of non-perfusion (NP), has a straightforward relationship with the risk of disease progression [10,11,12] and the incidence of diabetic macular edema (DME) [13]. Additionally, defining areas of NP enables targeted retinal photocoagulation at the periphery, which requires a smaller area of retinal ablation compared to classic panretinal photocoagulation [14]. In light of this knowledge, the visualization of peripheral vascular abnormalities by WF-OCTA, if comparably reliable to UWF-FA, may have a similar diagnostic value. Studies comparing WF-OCTA and UWF-FA in the context of diabetic retinopathy are summarized in Table 1.

The result analysis must take into consideration the following issues: differences in the obtained field of view, agreement between the examinations, sensitivity, and specificity in the detection of areas of NV.

Most recent studies employed the newly introduced Xephilio OCT-S1 (Canon, Tokyo, Japan) WF-OCTA and compared the obtained scans to those taken using UWF-FA by Optos [16,17] or conventional FA [18]. Comparisons with Optos, which provides a larger field of view than the Xephilio OCT S1, revealed high agreement between the exams in detecting the NVs in a small study by Bajka et al. [16]; however, a large study by Hamada et al. [17] proved the superiority of UWF-FA in detecting NVs due to the larger field of view and possible segmentation errors occurring with WF-OCTA. On the other hand, compared to standard FA, the Xephilio OCT S1 proved to be equally effective in detecting peripheral neovascularization [18]. One recent study used the Zeiss Plex Elite WF-OCTA montage to determine the utility of the non-perfusion index (NPI) in diagnosing severe non-proliferative retinopathy (NPDR) and proliferative diabetic retinopathy (PDR) [15]. Although NPI was significantly higher for PDR compared to NPDR, its correlation with the proliferative status of retinopathy was diminished as some of the NVs were outside the range of the WF-OCTA exam. The study’s conclusions are similar to those of Hamada et al.: detection of NV based on WF-OCTA is not as reliable as with UWF-FA due to the smaller field of view.

Older studies typically included only the Plex® Elite 9000, (Carl Zeiss Meditec Inc., Dublin, CA, USA) WF-OCTA with a mosaic of five images to obtain a large field of view [19,20,21,22,23,24]. Comparisons with Optos [19,20,22,23,24] revealed a similar sensitivity in detecting NVs and NPs and its superiority over the standard FA when OCTA was equipped with a plus 20 D lens that significantly enlarged the field of view [21]. Some authors reported high agreement between the UWF-FA and WF-OCTA devices in classifying DR [19]. Discrepancies between the two examinations were noted in older studies in the interpretation of visualized vascular lesions. Pellegrini et al. [21] defended classic FA, stating that it provides more reliable data on the perfusion status of the retina; Russell et al. [22] and Zhang et al. [24] stated that OCTA provides more details of the observed neovascular changes.

The introduction of WF-OCTA increased the detection rate of vascular abnormalities in DR compared to standard OCTA [25]. Both NP areas and the ischemic index were evaluated more precisely with 24 × 20 mm WF-OCT compared to 12 × 12 mm scans. In everyday practice, it must be remembered that the evaluation of the fundus cannot be omitted. The combination of WF-OCTA (prototype single capture at 65 degrees by Plex Elite) and UWF-CP can provide valuable information on retinal vasculature in DR [26]. Example of WF-OCTA imaging with 24 × 20 mm scans is provided in Figure 1. For comparison, the UWF-FA image of proliferative DR is shown in Figure 2. The field of view is definitely wider for the UWF Optos system.

WF-OCTA (24 × 20 mm) in DR can be used to quantify the vascular morphology at the peripheral retina in patients with DR [27]. Alterations in angio-OCT parameters at the peripheral retina are noted in vascular density and the thickness of vascular capillary complexes and can serve as predictors of DR development/progression. A similar approach can be employed with diabetic patients without developed retinopathy [28]. An interesting utilization of WF-OCTA was presented in a case series by Wright et al. [29]. The authors used this device to monitor PDR during pregnancy, when classic angiographic examinations are generally contraindicated.

The evaluation of the DR type can be enhanced not only by WF-OCTA but also by WF-OCT without the “angio” option [30]. The authors used 14 × 9 mm fovea-centered scans with additional 6 × 6 mm scans oriented at the periphery (Silverstone, Optos). WF-OCT imaging, enhanced by additional peripheral scans, visualized the relationship between the suspected lesion and the retinal surface and posterior hyaloid, and determined the diagnosis of neovascularization instead of intraretinal microvascular abnormalities (IRMAs). The use of UWF-directed OCT enabled the detection of NV in an additional 25% of eyes, thus changing their classification to PDR.

As indicated by the above analysis, WF-OCTA is becoming a valuable tool for the examination of peripheral vascular changes in DR. Due to the large field of view and dye-independence, it has advantages over standard FA. Moreover, the lack of dye leakage enables a more precise evaluation of the vascular network at the periphery, especially in relation to the neovascularization of the RPE and posterior vitreous. Nevertheless, UWF-FA with the largest available field of view, 200 degrees, remains the most reliable tool for detecting vascular peripheral pathologies such as NVs and NPs.

## 4. Retinal Vein Occlusion

UWF-FA has been proven to be a valuable tool in imaging peripheral retinal areas in RVO [31]. Its application resulted in the introduction of the ischemic index (ISI) formula, which expresses the relationship between non-perfused areas and the total retinal area imagined with WF scanning laser ophthalmoscopy [32,33,34,35]. ISI is based on the pixel count and is calculated automatically by some software. Higher ISI values are associated with a higher risk of developing NV in BRVO; Tsui et al. suggested a cut-off value of 45% in this respect [36]. The evaluation of UWF-FA and WF-OCTA showed high agreement between UWF-FA and standard OCTA in the evaluation of the extent of areas of non-perfusion. Existing studies on WF-OCTA in RVO are summarized in Table 2.

Examples of WF-OCTA and WF-OCT imaging in RVO is provided in Figure 3 and Figure 4. WF-OCTA enables the precise visualization of areas of non-perfusion at the peripheral retina and collateral vessels. For comparison, the UWF-FA image of RVO is provided in Figure 5. The visualized retinal area is wider for fluorescein angiography compared to OCTA.

A strong correlation between UWF-FA and WF-OCTA reflects the accurate detection of NP areas [37,39,41] and the ischemic index [38]. It must be emphasized that, as with DR, the area of visualized NP is significantly larger in the case of UWF-FA exams [39,41]. WF-OCTA is reportedly much more precise in detecting NVs compared to standard ophthalmoscopic examinations [40] or standard FA [42,43]. OCTA provides details of the morphology of neovascularization that cannot be detected using angiographic tests [40]. Such information is important in the context of the potential progression of NV and eligibility for pars plana vitrectomy (PPV).

Besides WF-OCT and WF-OCTA exams, standard OCTA is often used to evaluate vascular alterations in RVO. Tang et al. measured the area of the periarterial capillary free zone (CFZ) and the ratio of CFZ to the artery area after anti-VEGF treatment [44]. The study, based on 12 × 12 mm scans and performed with Plex® Elite 9000, (Carl Zeiss Meditec Inc., Dublin, CA, USA) revealed a significant decrease in areas of non-perfusion noted after treatment with intravitreal injections. OCTA also provides more detailed information on the development of collateral vessels, which are visualized with both a standard view and WF-OCTA [37].

## 5. Peripheral Retinal Degeneration

The search regarding peripheral retinal lesions and WF-OCT or WF-OCTA revealed four studies involving WF-OCT.

Govetto et al. retrospectively analyzed peripheral vitreoretinal interfaces with WF-OCT in cases with rhegmatogenous pathology [45]. The authors reported interesting findings concerning the relationship between the presence of specific types of peripheral retinal degeneration and the vitreoretinal interface status. The retrospective analysis revealed 166 lesions present in the observed cases: 106 horseshoe tears, 22 operculated, 30 non-operculated holes (OHs), six giant tears, and two peripheral lamellar defects. The posterior vitreous detachment (PVD) was present in all eyes with tears and OHs but in fewer eyes with non-OH (26.3%), *p* < 0.001. Axial traction at the tears was evident at their anterior border (100%) but not the posterior one (17%), *p* < 0.001. OHs located posterior to the vitreous base were free from vitreous traction and presented with a morphology similar to macular holes. On the other hand, non-OHs were located anteriorly with signs of tangential traction in 76.7% of cases. Peripheral vitreoschisis was prevalent in non-OHs (83.3%) but not in horseshoe tears (16%), *p* < 0.001. Horseshoe tears and non-OHs were more often associated with retinal detachment, 54.7% and 50%, respectively, compared to OHs (22.7%), *p* = 0.023. All these data shed new light on the pathogenesis of rhegmatogenous lesions and their risk of progression to retinal detachment. With WF-OCT, such an evaluation is possible and might help choose the best therapeutic option (e.g., protective laser photocoagulation). Figure 6 provides a WF-OCT scan of an operculated retinal hole located at the periphery.

A similar topic was investigated by Kurobe et al. [46]. The authors evaluated peripheral retinal degeneration and retinal detachment in 31 consecutive patients (37 eyes) using WF-OCT scans of 23 mm in width (Silverstone, Nikon Japan Healthcare, Tokyo, Japan). Lattice degeneration was found in eight eyes, paving stone degeneration in four eyes, unclassified in four eyes, retinal tears in twelve eyes (all horseshoe type), and retinal holes in nine eyes. The lesions were located at the mid-periphery (23 eyes) or far-periphery (14 eyes). Important findings were reported concerning the peripheral lesions. WF-OCT easily visualized the subretinal fluid, the detached edge of the degeneration, and vitreoretinal traction—factors that are important due to their potential progression. Rhegmatogenous retinal detachment was noted in 15 eyes, comprising one preoperative eye and 14 postoperative eyes. Vitreoretinal traction (VRT) was present in 10 of 27 eyes without a history of PPV. Similar imaging with WF-OCT scans of 23 mm in width was performed by Stanga et al. [47]. Both Kurobe and Stanga emphasized the ease of the implementation and reliability of WF-OCT imaging for peripheral retinal disorders. The imaging of VRT with WF-OCT is provided in Figure 7.

The presence and classification of PVD were evaluated with the use of WF-OCT (mosaic of images) by Tsukahara et al. [48]. The authors included 144 healthy eyes of patients aged 29–95 years in the study. They proposed the classification of PVD into five stages: stage 0, without PVD (two eyes, both aged 21 years); stage 1, peripheral PVD limited to paramacular to peripheral zones (88 eyes, mean age 38.9 ± 16.2 years); stage 2, perifoveal PVD extending to the periphery (12 eyes, mean age 67.9 ± 8.4 years); stage 3, peripapillary PVD with persistent vitreopapillary adhesion alone (seven eyes, mean age 70.9 ± 11.9 years); and stage 4, complete PVD (35 eyes, mean age 75.1 ± 10.1 years). All stage 1 PVDs were observed in the paramacular to peripheral region. Progressing to stage 2 PVD, the area of PVD extended in two directions: posteriorly to the perifovea and anteriorly to the periphery. Vitreoschisis was observed in 41.2% of eyes at PVD initiation. These observations with WF-OCT provided novel and precise information on the location and progression of PVD. Such data would have been difficult to obtain with standard OCT (limited field of view) or ultrasound examination (lower resolution). An example of PVD imaging with WF-OCT is provided in Figure 8.

Despite the limited material on the diagnosis of peripheral retinal lesions with WF-OCT, this study’s findings indicate that this diagnostic modality can play a crucial role in the follow-up of retinal degeneration and local conditions post-retinal surgery. The reliable detection of VRT and subretinal fluids with WF-OCT imaging can provide substantial information for the effective management of these disorders and decision making, especially concerning surgical procedures.

## 6. Other Clinical Entities

The use of WF-OCT enabled the determination of novel anatomic features of familial exudative vitreoretinopathy (FEVR). These included retinoschisis, focal retinal thickening, and the sudden thinning of the retina and retinal ridge. Additionally, UWF-SLO revealed a temporal mid-peripheral vitreoretinal interface abnormality (TEMPVIA), which was found in 88.3% of FEVR patients [49]. In another study on FEVR, WF-OCTA had superior performance compared to UWF-SLO and comparable performance to UWF-FA in detecting peripheral vascular abnormalities, avascular zones, neovascularization, and TEMPVIA [50].

WF-OCTA was reported as a valuable imaging modality for retinal racemose hemangioma [51]. Fluorescein angiography showed multiple lesions with intense leakage that obscured the view of the vessels. By contrast, WF-OCTA clearly presented the retinal capillaries of the hemangioma and adjacent retina.

WF imaging has the potential to diagnose ocular tumors located at the periphery. Attempts at such visualizations were made by McNabb et al. using a commercially available SS laser enhanced by additional indirect Volk lenses [52]. The authors achieved more than twice the enlargement of the field of view compared with a standard OCT device. The prototype WF-OCT system enabled the visualization of 15 out of 16 tumors with a single-scan acquisition in the primary gaze.

Extended field imaging, including UWF-FA and WF-OCTA (Plex® Elite 9000, Carl Zeiss Meditec Inc., Dublin, CA, USA) with +20.0 D lens, was employed in the management of choroidal melanoma after radiation therapy [53]. EFI OCTA provided a larger view of the areas compared to standard OCTA and a standard 55-degree fundus camera. The images showed a good definition of retinal and choroidal changes after radiotherapy, thereby enhancing the management of these patients.

## 7. Conclusions

Wide-field imaging with OCT and OCTA provides substantial information on retinal disorders involving the peripheral retina. Vascular diseases such as DR and RVO can be evaluated with WF-OCTA with results superior to standard fluorescein angiography. As long as convenience and reliability is considered, WF- OCTA imaging is on a good track to become an alternative to standard field-of-view FA imaging.

UWF-FA provides a significantly larger field of view compared to WF-OCTA and still has advantages regarding the evaluation of areas of NP and peripheral NV; however, detailed information on the vascular morphology of peripheral changes should be obtained via WF-OCTA and not angiographic examinations, which are characterized by dye leakage that obscures the view. It has to be remembered though that obtaining good-quality WF-OCTA images is laborious and typically requires many repetitions. Usually, such scans are burdened with artifacts that make precise analysis difficult, especially quantitative assessments. The improvement in scanning speed and elimination of artifacts by device software is required for WF-OCTA to become a widely used tool in clinical practice.

WF-OCT can serve as a valuable tool for the detection and evaluation of vitreoretinal traction, posterior vitreous detachment, and peripheral retinal degeneration, and guide therapeutic decisions concerning a patient’s eligibility for surgical procedures. It facilitates information on the retinal–vitreous interface that can be sometimes hard to obtain by a simple fundus examination. Contrary to WF-OCTA, it is a fast test that can be easily utilized in a busy clinical practice.

And the end of this review, it can be stated that the reliability and subsequent wide use of WF-OCT and WF-OCTA examinations is strongly linked to technological progress, which continuously enables higher scanning speeds and larger fields of view.

## Figures and Tables

**Figure 1 diagnostics-14-00321-f001:**
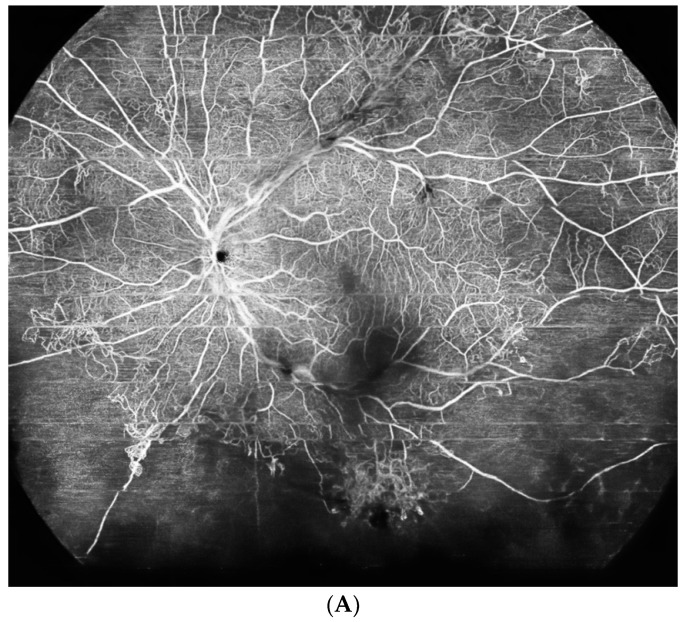
(**A**,**B**). Proliferative diabetic retinopathy on WF-OCTA. (Xephilio OCT-S1 (Canon, Tokyo, Japan)). (**A**): The RPE-choroid scan shows the vascular network at the superficial capillary plexus (SCP) with large areas of hypoperfusion and NVE. The shadowing on the scan results from the presence of vitreous hemorrhage. (**B**): The internal limiting membrane (ILM) slab shows NVs protruding to the vitreous.

**Figure 2 diagnostics-14-00321-f002:**
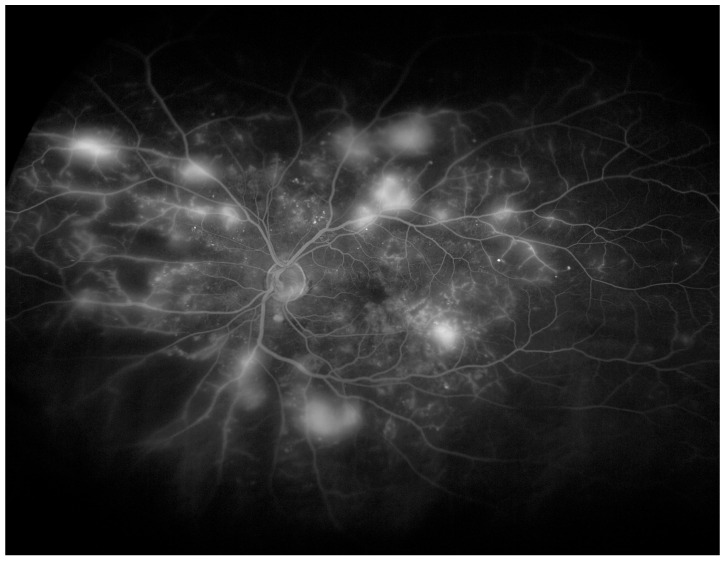
Imaging of proliferative diabetic retinopathy with UWF- FA (Optos^®^ 200Tx). Field of view provided with wide-field fluorescein angiography is considerably wider compared to Wf-OCTA from Figure 1.

**Figure 3 diagnostics-14-00321-f003:**
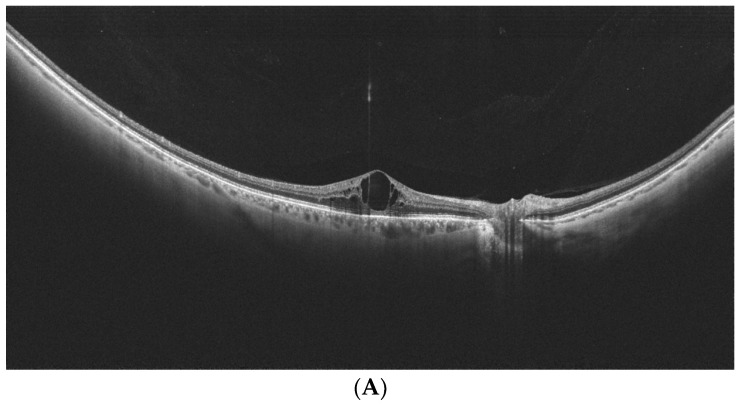
(**A**): The WF-OCT scan reveals cystoid macular edema. (**B**): WF-OCTA shows well-developed collateral vessels and minor areas of hypoperfusion in the course of BRVO. Xephilio OCT-S1 (Canon, Tokyo, Japan)).

**Figure 4 diagnostics-14-00321-f004:**
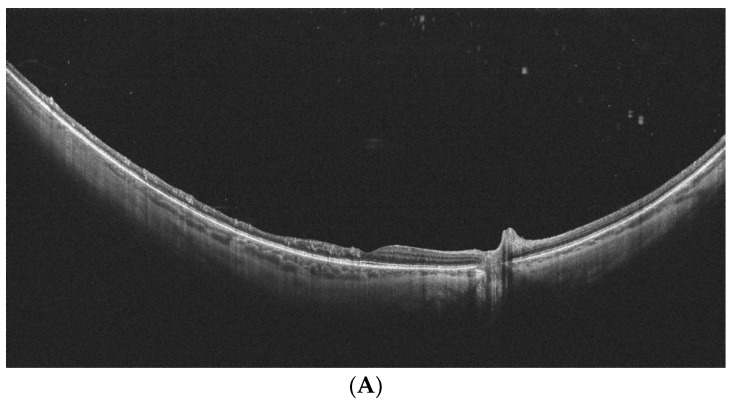
(**A**): The WF-OCT scan shows significant retinal thinning in the temporal part of the retina. (**B**): WF-OCTA shows large areas of hypoperfusion and arterio-venous anastomoses located in the inferotemporal sector of the retina (ischemic BRVO). Xephilio OCT-S1 (Canon, Tokyo, Japan).

**Figure 5 diagnostics-14-00321-f005:**
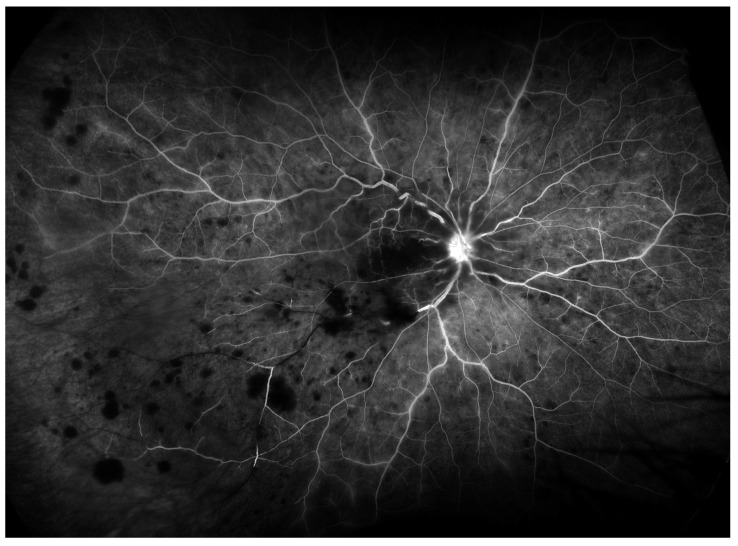
Imaging of RVO with UWF—FA (Optos^®^ 200Tx). Areas of hypoperfusion are located in inferotemporal quadrant of the retina. The field of view is wider than provided with WF-OCTA.

**Figure 6 diagnostics-14-00321-f006:**
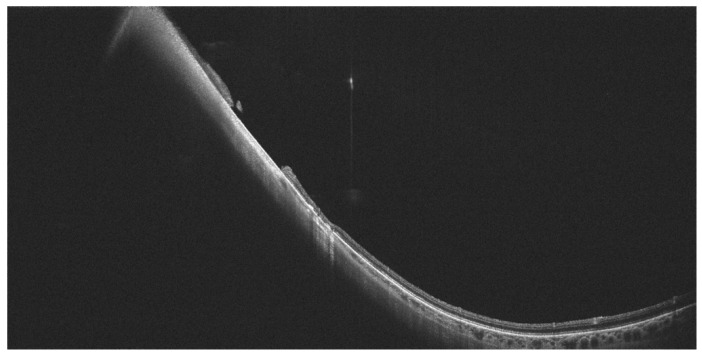
Peripheral operculated hole at the far periphery. Absence of retinal detachment/traction. (Xephilio OCT-S1, (Canon, Tokyo, Japan)).

**Figure 7 diagnostics-14-00321-f007:**
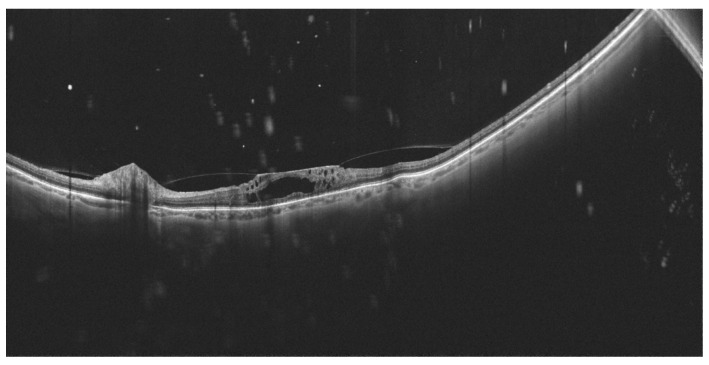
Vitreomacular traction visualized by WF-OCT. PVD is absent at the periphery. (Xephilio OCT-S1, (Canon, Tokyo, Japan).

**Figure 8 diagnostics-14-00321-f008:**
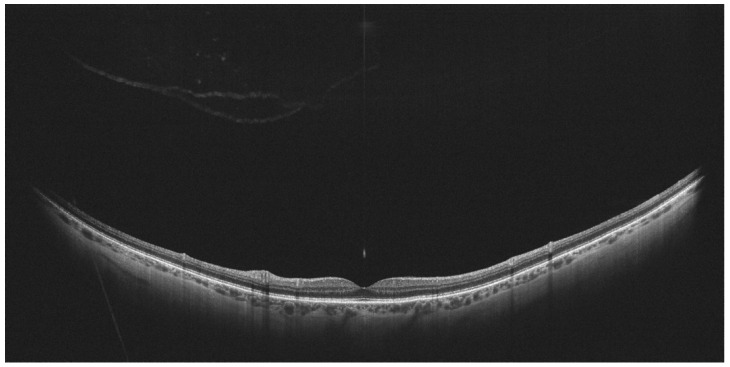
Complete posterior vitreous detachment on WF-OCT. (Xephilio OCT-S1, (Canon, Tokyo, Japan)).

**Table 1 diagnostics-14-00321-t001:** Comparison of studies on WF-OCTA versus wide-field and conventional fluorescein angiography in the context of diabetic retinopathy. Only the studies with a WF-OCTA field of view larger than 55 degrees were included.

Study	Equipment	Number of Eyes and Study Design	Results
Boile et al. 2023 [15]	UWF-FA Optos California (Optos, PLC, Dunfermline, Scotland) versus OCTA with montage of 12 × 12 mm fields of several visual fixations using a PLEX Elite 9000^®^ (field of 24 × 24 mm)	51 severe NPDR and PDR eyes, treatment-naïve; analysis of the utility of the non-perfusion index (NPI)	The NPI was significantly higher in eyes with PDR (18.94% vs. 7.51%; *p* < 0.01). Measurement of NPI on the whole wide-field OCTA image was not sensitive enough to replace the detection of NV for the diagnosis of PDR. UWF-FA detected NV foci outside the range of WF-OCTA.
Bajka et al. 2023 [16]	Xephilio OCT-S1 (Canon, Tokyo, Japan) (23 × 20 mm OCTA scan) versus Optos California (Optos, PLC, Dunfermline, United Kingdom) UWF-FA	15 eyes; evaluation of the retinal NP and NV; calculation of the ischemic index and VD	Both UWF-FA and WF-OCTA detected NP in 11 eyes (73%); NV was detected in four eyes by UWF-FA and in three eyes by WF-OCTA.
Hamada et al. 2023 [17]	UWF-FA (Optos California; Optos, PLC, Dunfermline, United Kingdom)) versus Canon Xephilio OCT-S1 (Canon, Tokyo, Japan) (23 × 20 mm) OCTA scan	108 gradable images; NV detection	With UWF-FA, 175 NV lesions were detected in 40 eyes; with WF-OCTA, 156 NV lesions were detected, with 118 of them confirmed by UWF-FA (true positive). Of 57 false negative lesions, the primary causes were being outside the scan range (26 lesions) and segmentation errors (21 lesions). WF-OCTA achieved a sensitivity of 95% and specificity of 88% in detecting eyes with NV.
Hirano et al. 2023 [18]	Xephilio OCT-S1 (Canon, Tokyo, Japan) (23 × 20 mm OCTA scan) versus conventional FA	64 eyes; NV detection in PDR patients	WF-OCTA revealed 96% (162) of NV sites when the scan was fovea-centered and 99% (166) when it was disc-centered compared to conventional FA (168 sites). With a mosaic of these two fields, all NVs were visualized by WF-OCTA.
Cui et al. 2021 [19]	UWF-FA (Optos California; Optos, PLC, Dunfermline, United Kingdom) versus OCTA with montage of 12 × 12 mm fields of several visual fixations using a Plex® Elite 9000, (Carl Zeiss Meditec Inc., Dublin, CA, USA)	152 eyes of 101 patients; comparing WF- OCTA with UWF-CP and UWF-FA in detection of DR lesions	WF-OCTA was superior to UWF-CP in detecting IRMAs (*p* < 0.001) and NVE/NVD (*p* = 0.007). The detection rates for microaneurysms, IRMAs, NVE/NVD, and NPs with WF-OCTA were comparable with those of UWF-FA (*p* > 0.05). A comparison of UWF-OCTA plus UWF-CP with UWF-FA showed identical detection rates for microaneurysms, IRMAs, NVE/NVD, and NP areas (*p* > 0.005). There was high agreement (κ = 0.916) between WF-OCTA and UWF-FA in classifying DR.
Picchi et al. 2020 [20]	UWF-FA Optos^®^ 200Tx (Optos, Dunfermline, United Kingdom) versus OCTA with 12 × 12 mm fields of five visual fixations using a Plex® Elite 9000, (Carl Zeiss Meditec Inc., Dublin, CA, USA)	82 eyes; NV detection in PDR patients	NVD was detected in 13 eyes by UWF-CP, 35 eyes by UWF-FA, and 37 eyes by WF-OCTA. Upon review of the 2500 OCT B-scans with superimposed flow overlay, NVD was confirmed in 37 eyes. UWF-CP analysis detected 62 foci of NVE out of the 196 confirmed by B-scan (31.6% detection rate). An additional 11 foci of NVE seen on UWF-CP were not confirmed by B-Scan (15% false positive rate). UWF-FA identified 182 foci of NVE (detection rate 91.3%); WF-OCTA detected 196 NVE (detection rate 100%). The rate of false positives for both UWF-FA and WF-OCTA was <2%. Respectively, the sensitivity and specificity of NVD detection were 35.1% and 97.8% for UWF-CP, 94.6% and 100% for UWF-FA, and 100% and 100% for WF-OCTA.
Pellegrini et al. 2019 [21]	55-degree FA (Spectralis Heidelberg Engineering, Heidelberg, Germany) versus standard OCTA Plex® Elite 9000, (Carl Zeiss Meditec Inc., Dublin, CA, USA) 12 × 12 mm and the same OCTA with prototype +20.0 D lens	43 eyes (32 DR, 8 RVO, 2 RVO with radiation retinopathy, and 1 Coats disease); comparison of field of view, NP areas, NV, and vessel density	The extension of NP areas was significantly larger with extended field imaging OCTA versus standard OCTA and FA (34.22 vs. 20.46 vs. 27.56 mm^2^). No differences were found between the devices with respect to the detection of NV. The mean vessel density was significantly lower with extended field imaging OCTA. Nevertheless, FA gave more details of the perfusion status of the retina; in some cases, OCTA erroneously imagined hypoperfused areas that were not confirmed by FA.
Russell et al. 2019 [22]	UWF-FA Optos^®^ 200Tx Optos, Dunfermline, United Kingdom) versus Plex® Elite 9000, (Carl Zeiss Meditec Inc., Dublin, CA, USA) OCTA with 12 × 12 mm fields of five visual fixations	20 eyes of 15 patients; observation of NV after panretinal photocoagulation (PRP) at one week, one month, and three months	The en-face SS-OCTA 12 × 12 mm vitreoretinal interface slab images showed NV at baseline in 18 of 20 eyes (90%). Concerning the remaining two eyes, the posterior pole montage captured peripheral NV in one eye; in the other eye, no evidence of NV was detected with either UWF FA or SS-OCTA. Based on UWF-FA, eight eyes (47%) progressed and nine eyes (53%) regressed. Identical conclusions were reached from SS-OCTA scans. SS-OCTA provided a more detailed visualization of the vascular changes.
Sawada et al. 2018 [23]	UWF-FA Optos^®^ 200Tx Optos, Dunfermline, United Kingdom versus OCTA with 12 × 12 mm fields of five visual fixations using a Plex® Elite 9000, (Carl Zeiss Meditec Inc., Dublin, CA, USA)	58 eyes; detection of NP areas or NV	NP areas were detected in 47 eyes by UWF-FA versus 48 eyes by OCTA; NV was detected in 25 eyes by UWF-FA versus 26 by OCTA. The sensitivity for the detection of NP areas using OCTA was 0.98, and the specificity was 0.82; the sensitivity for the detection of NV was 1.0, and the specificity was 0.97.
Zhang et al. 2018 [24]	Plex® Elite 9000, (Carl Zeiss Meditec Inc., Dublin, CA, USA) montage of sixteen 6 × 6 mm images (100 degrees) versus traditional 50-degree FA	Three PDR patients; comparison of montage UWF OCTA to 50-degree FA	More details regarding the capillary network and visualization of NVs were missing in standard FA.

DR: Diabetic retinopathy; PDR: proliferative diabetic retinopathy; RVO: retinal vein occlusion; WF-OCTA: wide-field OCTA; UWF-CP: ultra-wide-field color photography; UWF: ultra-wide-field; FA: fluorescein angiography; NVD: neovascularization at disc; NVE: neovascularization elsewhere; NP: non-perfusion; NV: neovascularization; VD: vessel density; PRP: panretinal photocoagulation; SS: swept-source; NPI: non-perfusion index.

**Table 2 diagnostics-14-00321-t002:** Studies evaluating UWF-OCTA (visualization of at least 90 degrees) in the diagnostics of RVO.

Study	Equipment	Number of Eyes	Study Design and Results
Siying et al. 2022 [37]	UWF-FA Optos^®^ 200Tx (Optos, Dunfermline, United Kingdom) versus a single capture of 24 × 20 mm wide-field SS-OCTA scan using BM400K (Bmizar, TowardPi Medical Technology Co., Ltd., Beijing, China)	32 treatment-naïve eyes	Comparison of the FAZ area, FAZ perimeter, and NP areas (measurements at SCP); the measurements were performed manually. The median FAZ area was 0.373 mm^2^ (range 0.277–0.48) on SS-OCTA and 0.370 mm^2^ (range 0.277–0.48) on UWF-FA. The median FAZ perimeter was 2.480 mm (range 2.011–2.998) and 2.330 mm (range 2.027–2.807) on the SS-OCTA and UWF-FA images, respectively. No significant difference was noted (*p* = 0.818 and *p* = 0.536, respectively). The mean NP area was larger on SS-OCTA than on UWF-FA (89.977 ± 78.805 vs. 87.944 ± 77.444 mm^2^, *p* = 0.037) for corresponding images; SS-OCTA was superior in visualizing capillary changes and collateral vessels.
Glacet-Bernard et al. 2021 [38]	WF-OCTA Plex® Elite 9000, (Carl Zeiss Meditec Inc., Dublin, CA) with a montage of five 12 × 12 mm images versus UWF-FA Optos^®^ 200Tx (Optos, Dunfermline, United Kingdom	43 eyes	The ischemic index on UWF-FA and the vascular density in the superficial and deep plexus correlated significantly (*p* = 0.019, r = 0.357 and *p* < 0.013, r = 0.375, respectively). The qualitative classification on wide-field OCTA and the ischemic index on UWF-FA correlated significantly (*p* < 0.001, r = 0.618). For the detection of marked non-perfusion (ischemic index ≥ 25%), wide-field OCTA had a sensitivity of 100% and a specificity of 64.9%.
Kadamodo et al. 2021 [39]	Plex® Elite 9000, (Carl Zeiss Meditec Inc., Dublin, CA) montage of five 12 × 12 mm scans versus single OCTA scan and UWF-FA, Optos^®^ 200Tx (Optos, Dunfermline, United Kingdom	26 treatment-naïve eyes	The retinal areas of NP measured using single OCTA and panoramic OCTA were compatible with those measured using UWF-FA (*p* < 0.001 for both). Retinal neovascularization lesions were observed in 4 (15.4%) of 26 eyes. For patients with accompanying neovascularization, the retinal NP measured using UWF-FA, single OCTA, and panoramic OCTA were 187.9 ± 39.5 mm^2^ (disc area 109.9 ± 21.4), 34.3 ± 13.7 mm^2^ (disc area 19.9 ± 7.7), and 106.6 ± 24.5 mm^2^ (disc area 62.4 ± 13.6), respectively, and were larger than for those without neovascularization (*p* < 0.001, *p* < 0.014, and *p* < 0.001, respectively).
Huemer et al. 2021 [40]	Plex® Elite 9000, (Carl Zeiss Meditec Inc., Dublin, CA) montage of five 12 × 12 mm scans versus standard examination (visual acuity and fundus evaluation on biomicroscopy) and UWF-FA, Optos^®^ 200Tx (Optos, Dunfermline, United Kingdom, if available	39 eyes with ischemic RVO	Retrospective study. NVE was detected in 41% of eyes by WF-OCTA versus 10.3% via standard examination; in one case, NVE detected by OCTA was not revealed by UWF-FA. WF-OCTA provided sea-fan and nodular morphological characteristics of NVE vessels. UWF-OCT images provided details on the location of NV in reference to the posterior hyaloid (sea-fan growing along the posterior hyaloid and nodular close to the retinal surface). Nodular vessels were not detected during standard examinations but only via UWF exams. Sea-fan vessels were detected in all cases during standard exams.
Shiraki 2019 [41]	Plex® Elite 9000, (Carl Zeiss Meditec Inc., Dublin, CA) a mosaic of five 12 × 12 mm scans versus UWF-FA Optos^®^ 200Tx (Optos, Dunfermline, United Kingdom	23 eyes	The mean area of NP in the OCTA images was 81.0 ± 66.8 mm^2^ (range 0.0–188.8) versus 84.7 ± 72.5 mm^2^ (range 0.0–221.9) on FA (for corresponding examined retinal areas in both tests). The total area of NP on FA had a mean of 184.1 ± 167.7 mm^2^. The mean VD was 27.6 ± 3.5% (range 19.6–33.7); the mean VL was 12.4 ± 8.5% (range 5.4–31.3). Separate regression analysis of the areas of retinal non-perfusion in FA (*p* = 0.0004, *R*^2^ = 0.4627) and the total FA (*p* = 0.0008, *R*^2^ = 0.4214) images showed a significant association with the VL.
Kakihara et al. 2018 [42]	Plex® Elite 9000, (Carl Zeiss Meditec Inc., Dublin, CA) with EFI (+ 20 D lens); 12 × 12 mm scans versus standard FA	23 eyes of 22 patients	The average extension rate of EFI-SS-OCTA over SS-OCTA was 1.39 ± 0.06, and the average scanning area was enlarged by 76.4%. There was a moderate concordance to FA images in reference to NP areas (Cohen’s unweighted Kappa coefficient = 0.60). The OCTA images showed a larger extent of NP compared to FA, as the authors note, due to a lack of masking by the leakage from retinal vessels present in FA.
Kimura et al. 2016 [43]	RTVue XR Avanti OCT with AngioVue^®^ with extended field imaging (+20 D lens), basic 8 × 8 mm scan versus standard FA (Heidelberg Retina Angiograph 2, Heidelberg, Germany) with 30 degrees of field of view	10 eyes of nine patients	Enlargement of 188.5% of the standard field with +20 D lens; good definition of NP areas in SCP. The average area of NP determined by EFI OCTA was 18.3 mm^2^ versus 16.8 mm^2^ using fluorescein angiography.

RVO: Retinal vein occlusion; NP: non-perfusion; SCP: superficial capillary plexus; UWF-FA: ultra-wide-field fluorescein angiography; NVE: neovascularization elsewhere; VD: vessel density; VL: vessel length; FAZ: foveal avascular zone; EFI: extended field imaging; SCP: superficial capillary plexus.

## Data Availability

Not applicable.

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
