# Peer review of "Advantages of the Utilization of Wide-Field OCT and Wide-Field OCT Angiography in Clinical Practice"

_diagnostics, 2024, doi:10.3390/diagnostics14030321_

Round 1

Reviewer 1 Report

Comments and Suggestions for Authors

Authors performed huge research.

I suggest to exclude paragraph 5 "Peripheral retinal telangiectasia and Coats disease", because the topic is "Advantages of the utilization of wide-field OCT and wide-field OCT angiography in clinical practice". So, if authors didn`t find information about utiliztion in clinical practice better not include it.

Also this paragaph does not contain practical information and may be excluded: "WF-OCTA and WF-OCT have the potential to become useful diagnostic tools for diseases linked with peripheral vascular changes. These include von Hippel–Lindau disease, which is characterized by the incidence of retinal capillary hemangioma [61–62], and the rare Susac syndrome, in which peripheral vascular changes include arteriolitis, BRVO, and focal ischemia [63]. At the time of our study, there were no reports published on these diseases." 

In Table 2. Huemer et al. 2021 mentioned “standard examination” – what does it mean – biomicrophthalmoscopy with Goldman lens or 90D lens or indirect ophthalmoscopy?

Reviewer 2 Report

Comments and Suggestions for Authors

The authors have presented a descriptive and systematic analysis of the advantages associated with wide-field Optical Coherence Tomography (OCT) and wide-field OCT Angiography (OCTA) examinations when compared to established angiographic imaging modalities such as standard Fluorescein Angiography (FA), Ultra-Widefield FA (UWF-FA), and Ultra-Widefield Color Photography (UWF-CP). The information provided is comprehensive and well-organized; However, there are some problems to be addressed.

The term "retinal periphery" is frequently used in this review. Would "peripheral retina" be a more appropriate and precise expression?

1.       On page 2, the statement, "Diabetic retinopathy (DR) is a vascular disorder often associated with peripheral pathologies that influence its classification, progression, and the development of diabetic macular edema," introduces the term "peripheral pathologies," which may be potentially confusing. Consider refining or clarifying this term for improved precision.

2.       Table 1 appears to lack a well-structured format, impeding ease of readability.

3.       In Table 1, the font in the results cell for Russell et al. 2019 [22] differs from that of other entries.

4.       Figures, except for Figure 4, do not seem to be explicitly referenced in the text, despite having figure legends. Integrating references to these figures within the narrative would enhance overall coherence.

5.       Notably, there is no dedicated discussion section. Consider including a discussion or expanding the conclusion.

6.       The authors may consider including FA images to facilitate a more visual comparison.
